# Understanding Weak-to-Strong Generalization: A Spectral Analysis

## Abstract

Weak-to-Strong (W2S) generalization, where a student model surpasses its weaker teacher using the teacher's labels, is widely studied recently. We theoretically analyze this problem using a kernel ridgeless regression student in a Reproducing Kernel Hilbert Space (RKHS), learning from a teacher with systematic bias and output variance. Our derived asymptotic bias-variance decomposition reveals how teacher errors are processed by the student. This processing is critically mediated by the student's kernel eigenvalues and, crucially, its choice of operational modes and their alignment with the teacher's signal. We then elucidate precise conditions for W2S: outperformance hinges on this selection effectively managing the trade-off between bias and variance. Such strategic mode utilization can lead to a more favorable bias configuration via selectively ignoring the teacher's biased modes, or a reduction of teacher variance through modes with beneficial eigenvalue properties. Our experiments validate these theoretical conditions, demonstrating successful W2S generalization and underscoring the critical impact of kernel selection on navigating the bias-variance trade-off.

## 1 Introduction

The rapid advancements in machine learning have produced models, notably Large Language Models (LLMs) (Vaswani et al., 2017; Brown et al., 2020), that exhibit remarkable capabilities across a range of complex tasks. This progress, however, introduces a significant operational challenge: how can we effectively supervise, align, or further enhance these powerful models when the available supervisory signals, whether from human annotators, existing datasets, or other automated systems, are inherently weaker or less proficient than the student model itself? As our models become more intelligent, our ability to provide perfect guidance diminishes. The paradigm of Weak-to-Strong (W2S) generalization, as formulated by Burns et al. (2023), directly confronts this increasingly common scenario. It investigates the intriguing, and perhaps counter-intuitive, possibility that a more capable student model, when trained solely on labels generated by a demonstrably weaker teacher, can not only learn from this imperfect guidance but also ultimately generalize to a performance level that significantly exceeds that of its teacher. Elucidating the principles and conditions that enable such outperformance is crucial for the continued development of sophisticated AI systems, for building trust in their capabilities, and for devising more efficient methods for AI alignment and improvement.

To theoretically analyze W2S generalization, we turn to kernel methods and the framework of Reproducing Kernel Hilbert Spaces (RKHS) (Schölkopf and Smola, 2002; Wahba, 1990). RKHS theory, particularly Mercer's theorem (Mercer, 1909), allows for a spectral decomposition of kernel functions. This spectral perspective is useful for W2S analysis as it provides a clear way to understand how a student model, characterized by its kernel and associated modes of operation, processes the signal and noise from a teacher. It enables a detailed investigation into how the student's kernel structure influences its ability to learn from, and potentially filter or correct, the imperfect supervision provided.

In this study, we model the student as a kernel ridgeless regression (KRR) estimator within an RKHS. The student learns from a weak teacher that introduces both a systematic bias $b(x)$ (relative to the true function $f_*(x)$) and output variance $\sigma^2$. We concentrate on the ridgeless setting due to its connections with the behavior of overparameterized models and phenomena like benign overfitting Bartlett et al.

(2020). Our aim is to use a spectral analysis to determine the conditions under which this KRR student can generalize beyond its weaker teacher.

Our contributions are as follows: First, we establish a theoretical framework for analyzing a KRR student trained on outputs from a systematically biased and noisy weak teacher. This includes deriving the student's asymptotic bias-variance decomposition. Second, building on this decomposition, we elucidate precise conditions under which W2S generalization can occur. We show that the student's success can stem from achieving a more favorable bias configuration by effectively filtering teacher bias in certain modes, or by substantially reducing the impact of the teacher's variance. Third, we discuss the stringent nature of this variance reduction requirement for KRR models. We also explore the nuanced bias-variance trade-off involved in the student's selection of its effective operational modes determined by its kernel, offering insights for kernel selection aimed at optimizing W2S performance, particularly when true validation data is scarce. Finally, our theoretical analysis, including the impact of kernel and mode selection on the bias-variance trade-off, is corroborated by numerical experiments demonstrating W2S generalization.

## 2    RELATED WORK

Weak-to-strong (W2S) generalization refers to the surprising phenomenon where a strong student model, when trained on labels or feedback from a weaker teacher model, achieves better test performance than the teacher itself. This scenario was first highlighted by Burns et al. (2023), who showed that large language models fine-tuned on the coarse outputs of smaller models can surpass the teacher on downstream tasks. Subsequent empirical studies have observed W2S generalization across a range of settings, including vision models and other modalities. For example, in computer vision, a high-capacity model pretrained on image data (e.g. a Vision Transformer) can be fine-tuned on labels predicted by a weaker CNN and still outperform that CNN on ground truth labels (Guo and Yang, 2024). In NLP, W2S has been demonstrated in instructing large language models with smaller ones' generations (Burns et al., 2023), and even multi-teacher or iterative schemes have yielded W2S gains (Liu and Alahi, 2024). Following the initial W2S findings, researchers have proposed methods to enhance W2S generalization, such as improved loss functions and multi-teacher training to better align student and teacher predictions (Liu and Alahi, 2024), or data filtering and confidence-weighting strategies to let the student focus on reliable teacher outputs (Guo and Yang, 2024). W2S generalization has even been analyzed in controlled synthetic settings: Charikar et al. (2024) define metrics like performance gap recovery to quantify how much of the weak teacher's performance deficit a strong student can regain, finding significant gains in practice. These converging studies establish that W2S is a robust phenomenon across language, vision, and synthetic tasks, although certain pitfalls (e.g. "W2S deception" (Yang et al., 2024) when a student overfits the teacher's biases) have also been noted, underscoring the need for a deeper theoretical understanding of when and why W2S succeeds.

Recent work has begun to develop a theoretical foundation for W2S generalization. Lang et al. (2024) provide an analysis of W2S in a classification setting, introducing the notions of pseudolabel correction, where the student learns to correct the teacher's systematic errors, and coverage expansion, where the student generalizes to inputs where the teacher abstained or was uncertain. Their generalization bounds formalize the intuition that a strong student will outperform a weak teacher if it cannot perfectly mimic the teacher's mistakes without incurring additional error. In the regression setting, Dong et al. (2025) propose a framework based on the intrinsic dimension of model feature spaces: they show that W2S can occur when the student's effective dimensionality is lower than the teacher's and the student's representation aligns well with the teacher's so that the student captures the relevant signal but averages out the teacher's noise. This leads to a variance reduction explanation of W2S, wherein the student achieves a lower variance on the true target function by virtue of its higher capacity, more compact feature space. Complementarily, Ildiz et al. (2024) analyze a high dimensional linear distillation model and derive precise scaling laws for W2S: their results indicate that while a student can indeed outperform its teacher, the improvement in error rate may be bounded by the teacher's own learning dynamics (i.e. a student cannot circumvent fundamental data requirements set by the teacher's signal-to-noise ratio). Another line of analysis focuses on how representational differences enable W2S. For instance, Xue et al. (2025) study the role of feature quality and show that if the student's hypothesis class contains functions that better approximate the true target than any function the teacher can represent, then given enough data the student will correct

the teacher's errors on the harder parts of the input distribution. In comparison to these studies, our work offers a novel perspective by analyzing W2S in the context of kernel ridge regression (KRR). We derive an exact asymptotic bias–variance decomposition for a KRR student trained on a weak teacher's outputs, which allows us to pinpoint how the teacher's bias and variance are inherited or mitigated by the student. Our results bridge W2S generalization with classical learning theory tools, providing clarity on the conditions under which a student can outperform its teacher.

Finally, our approach is related to the broader literature on kernel methods and their generalization properties, especially in over-parameterized regimes. Weak-to-strong generalization also connects naturally with classic and modern results on kernel methods. Early RKHS theory formalized how a kernel's eigenvalues control bias–variance trade-offs (Schölkopf and Smola, 2002; Wahba, 1990). Recent work on the ridgeless regime shows that kernel ridge regression can still generalize benign overfitting when the spectrum decays sufficiently fast and the target is smooth (Belkin et al., 2019; Bartlett et al., 2020). Spectral analyses of learning curves further reveal how error concentrates on high-frequency modes and predict the familiar double-descent shape (Canatar et al., 2021). Our study leverages eigenfunction tools to show how a kernel student can preserve the weak teacher's low frequency signal while attenuating its noise.

## 3 NOTATIONS

Throughout this paper, we denote the input space by $\mathcal{X}$ and the output space by $\mathcal{Y}$. The probability distribution governing the inputs $x \in \mathcal{X}$ is $P(x)$, and $dP(x)$ will denote integration with respect to this measure. The Hilbert space of functions $f : \mathcal{X} \to \mathbb{R}$ that are square-integrable with respect to $P$ is denoted by $L^2(P)$, equipped with the inner product $\langle f, g \rangle_{L^2(P)} = \int_{\mathcal{X}} f(x)g(x)dP(x)$ and the corresponding norm $\|f\|_{L^2(P)}$. Standard mathematical operators for expectation, variance, and trace are denoted by $\mathbb{E}[\cdot]$, $\mathrm{Var}(\cdot)$, and $\mathrm{Trace}(\cdot)$, respectively. The $n \times n$ identity matrix is $I$.

We denote a symmetric, positive semi-definite kernel function as $K(x, x')$. This kernel defines a unique Reproducing Kernel Hilbert Space (RKHS), denoted by $\mathcal{H}_K$, with its associated inner product $\langle \cdot, \cdot \rangle_{\mathcal{H}_K}$ and norm $\| \cdot \|_{\mathcal{H}_K}$. Associated with the kernel $K$ and the data distribution $P(x)$ is the integral operator $T_K f(x) = \int k(x, x')f(x')dP(x')$. The population eigenvalues of $T_K$ are denoted by $\lambda_k \geq 0$, and the corresponding population eigenfunctions, which form an orthonormal basis for $L^2(P)$, are denoted by $\phi_k(x)$.

## 4 PROBLEM FORMULATION

### 4.1 ASSUMPTIONS

Our analysis relies on the following key assumptions:

**Assumption 4.1** (Data Generation). The input points $x_j \in \mathcal{X}$ are drawn i.i.d. from $P(x)$. The corresponding labels $y_j \in \mathbb{R}$ are direct outputs of a weak teacher model. We define the mean of the weak teacher's output as $f_T(x) = f_*(x) + b(x)$, where $b(x)$ is the systematic bias of the teacher's mean output. The student's training labels are thus $y_j = f_T(x_j) + \xi_j$, where $\xi_j = y_j - f_T(x_j)$ represents the teacher's deviation from its mean.

**Assumption 4.2** (Noise Properties from Teacher Variability). We assume $\mathbb{E}[\xi_j|x_j] = 0$ and $\mathrm{Var}[\xi_j|x_j] = \sigma^2$.

**Assumption 4.3** (Function and Mode Properties). The true function $f_*(x)$, the teacher's mean signal $f_T(x)$, and its bias $b(x)$ are assumed to belong to $L^2(P)$. The student model utilizes a kernel $K_S(x, x') = \sum_{k \in S_S} \lambda_{S,k} \phi_k(x)\phi_k(x')$, where $S_S$ is the set of mode indices supported by the student's kernel, and $\lambda_{S,k} > 0$ are the corresponding eigenvalues. The teacher's mean signal $f_T(x)$ is assumed to have its spectral support within a set of modes $S_W$, meaning $f_T(x) = \sum_{k \in S_W} \langle f_T, \phi_k \rangle_{L^2(P)} \phi_k(x)$. The noise $\xi(x)$ is also assumed to have its spectral content primarily within $S_W$.

**Assumption 4.4** (Kernel and Student Model). The student's kernel $K_S(x, x')$ is symmetric, continuous, and positive semi-definite, satisfying Mercer's theorem conditions, defining an RKHS $\mathcal{H}_{K_S}$. The student estimator $\hat{f}(x)$ is obtained via Kernel Ridge Regression (KRR) by minimizing the regularized

empirical risk:

$$\hat{f} = \arg\min_{h \in \mathcal{H}_{K_S}} \left( \frac{1}{n} \sum_{j=1}^{n} (y_j - h(x_j))^2 + \lambda_{reg} \|h\|_{\mathcal{H}_{K_S}}^2 \right)$$

We focus on the ridgeless setting ($\lambda_{reg} \to 0$), where the solution involves $K_n^\dagger$, the Moore-Penrose pseudoinverse of the empirical Gram matrix $(K_n)_{jl} = K_S(x_j, x_l)$.

Our goal is to analyze the expected squared error of this student estimator with respect to the true function $f_*(x)$, denoted $ER(\hat{f}) = \mathbb{E}[\|\hat{f} - f_*\|_{L^2(P)}^2]$. This formulation allows us to investigate how discrepancies in modal capabilities ($S_S$ vs. $S_W$) and the teacher's characteristics ($b(x), \sigma^2$) influence the student's ability to generalize to $f_*$.

# 5 BIAS-VARIANCE DECOMPOSITION AND ASYMPTOTIC ANALYSIS

## 5.1 FINITE-$n$ CONDITIONAL BIAS AND VARIANCE

For a fixed training set of $n$ input points, $X = \{x_j\}_{j=1}^{n}$, we first define the components of the student's error. The student estimator, averaged over the label noise $\xi$ but conditional on $X$, is $\hat{f}(x|X) = \mathbb{E}_\xi[\hat{f}(x|X, y)]$. For kernel ridgeless regression, this is given by:

$$\hat{f}(x|X) = S_n^*(X) K_n^\dagger(X) f_T(X)$$

where $f_T(X) = [f_T(x_1), \dots, f_T(x_n)]^T$. In terms of the student's kernel spectral components, this can be expressed as

$$\hat{f}(x|X) = \sum_{k \in S_S} \lambda_{S,k} (\Phi_k(X)^T K_n^\dagger(X) f_T(X)) \phi_k(x),$$

where $\Phi_k(X) = [\phi_k(x_1), \dots, \phi_k(x_n)]^T$.

The squared bias of the student estimator, conditional on $X$ and relative to the true function $f_*(x)$, is:

$$\text{Bias}_*^2(\hat{f}|X) = \|\hat{f}(\cdot|X) - f_*\|_{L^2(P)}^2$$

This term measures the systematic error of the noise-averaged student estimator for the given $X$, reflecting how this learned function aligns with $f_*$.

The variance of the student estimator $\hat{f}$ around its conditional mean $\hat{f}_0(\cdot|X)$, due to the teacher's output variability $\xi$ (with integrated variance $\sigma^2$), is:

$$\text{Var}_\xi(\hat{f}|X) = \mathbb{E}_\xi \left[ \|\hat{f}(\cdot|X, y) - \hat{f}(\cdot|X)\|_{L^2(P)}^2 \right]$$

This conditional variance can be shown as:

$$\text{Var}_\xi(\hat{f}|X) = \sigma^2 \text{Tr}(M(X)(K_n^\dagger(X))^2)$$

where the $n \times n$ matrix $M(X)$ has entries $(M(X))_{jl} = \langle K_S(x_j, \cdot), K_S(x_l, \cdot) \rangle_{L^2(P)}$. Using the Mercer decomposition of $K_S$, this can also be written as $\text{Var}_\xi(\hat{f}|X) = \sigma^2 \sum_{s \in S_S} \lambda_{S,s}^2 (\Phi_s(X)^T (K_n^\dagger(X))^2 \Phi_s(X))$. These conditional terms form the basis for the subsequent asymptotic analysis.

## 5.2 STUDENT'S ASYMPTOTIC BEHAVIOR

In the asymptotic limit ($n \to \infty$), the student's average prediction, $\mathbb{E}_\xi[\hat{f}(x)]$, converges to the projection of the teacher's mean signal $f_T(x)$ onto the subspace spanned by the student's active modes $\{\phi_k\}_{k \in S_S}$. Since $f_T(x)$ itself is supported only on modes $S_W$, this projection becomes $P_{S_S}(f_T)(x) = \sum_{k \in S_S \cap S_W} \langle f_T, \phi_k \rangle_{L^2(P)} \phi_k(x)$. Let us denote this projected teacher signal as $f_{S_S \cap S_W}(x)$.

The asymptotic squared bias of the student with respect to $f_*$ is then given by the squared $L^2(P)$ distance between this learned component of the teacher's signal and the true function:

$$\text{Bias}_S^2 = \|f_{S_S \cap S_W} - f_*\|_{L^2(P)}^2$$

$$\to \left\| \sum_{k \notin S_S \cap S_W} \langle f_* + b, \phi_k \rangle_{L^2(P)} \phi_k(x) + \sum_{k \in S_S \cap S_W} \langle b, \phi_k \rangle_{L^2(P)} \phi_k(x) \right\|_{L^2(P)}^2$$

$$= \sum_{k \notin S_S \cap S_W} |\langle f_* + b, \phi_k \rangle_{L^2(P)}|^2 + \sum_{k \in S_S \cap S_W} |\langle b, \phi_k \rangle_{L^2(P)}|^2.$$

For the asymptotic variance, when the student uses its kernel $K_S$ to learn from $f_T(x) + \xi(x)$, the noise that effectively propagates through the student's learning mechanism is the component of $\xi(x)$ that lies in $S_S \cap S_W$. Therefore, the student's asymptotic integrated variance due to this noise becomes:

$$Var_S \to \sigma^2 \sum_{k \in S_S \cap S_W} \frac{1}{\lambda_{S,k}}$$

This sum is now over the modes common to both the student's kernel support and the teacher's support, provided the sum converges. Therefore, the population expected risk becomes

$$ER(\hat{f})_S \to \sum_{k \notin S_S \cap S_W} |\langle f_* + b, \phi_k \rangle_{L^2(P)}|^2 + \sum_{k \in S_S \cap S_W} |\langle b, \phi_k \rangle_{L^2(P)}|^2 + \sigma^2 \sum_{k \in S_S \cap S_W} \frac{1}{\lambda_{S,k}}. \quad (1)$$

### 5.3 Conditions for Weak-to-Strong Generalization with Discrepant Modes

The weak teacher model has an expected error $ER(y) = \|f_T - f_*\|_{L^2(P)}^2 + \sigma^2$. W2S generalization occurs if $ER(\hat{f}) < ER(y)$, which translates to:

$$\sum_{k \notin S_S \cap S_W} |\langle f_* + b, \phi_k \rangle_{L^2(P)}|^2 + \sum_{k \in S_S \cap S_W} |\langle b, \phi_k \rangle_{L^2(P)}|^2 + \sigma^2 \sum_{k \in S_S \cap S_W} \frac{1}{\lambda_{S,k}} < \|b\|_{L^2(P)}^2 + \sigma^2.$$

First, W2S can occur via bias improvement. This happens if the student's effective mean prediction $f_{S_S \cap S_W}(x)$ is a better approximation of $f_*(x)$ than the teacher's full mean signal $f_T(x)$ is. This scenario can arise if the teacher's mean signal $f_T(x)$ contains erroneous components (i.e., parts of its bias $b(x)$) in modes that are within $S_W$ but outside of $S_S$ (modes in $S_W \setminus S_S$). By not having these modes, the student effectively filters out these specific erroneous components of $f_T$, leading to a mean prediction $f_{S_S \cap S_W}$ that is closer to $f_*$. This is a case where the student's "ignorance" of certain teacher modes is beneficial.

Second, W2S can occur through variance reduction. This requires:

$$\sum_{k \in S_S \cap S_W} \frac{1}{\lambda_{S,k}} < 1.$$

It implies that the student must be highly efficient at denoising the teacher's variability specifically within the functional subspace where they both operate and where the teacher provides its primary signal. This is achievable when the intersection is small enough.

Therefore, when student and teacher operate with different effective sets of modes, W2S generalization is possible if the student's modal structure allows it to either filter out specific bias components of the teacher or achieve a more effective suppression of the teacher's variance in the modes relevant to the learning task.

### 5.4 The Nature of the Trade-off in Student Mode Selection

Let's consider the contribution of a single, arbitrary mode $\phi_k$ to the student's total asymptotic error $ER(\hat{f})$. If the student does not include mode $k$ in its active set $S_S$ (effectively predicting zero for this component), its contribution to the total error from this mode is $(f_k^* + b_k)^2$, where $f_k^* = \langle f_*, \phi_k \rangle_{L^2(P)}$,

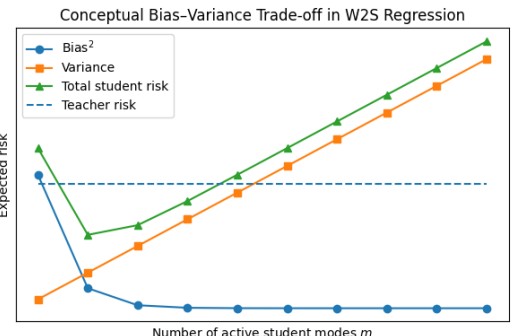

Figure 1: Intuitive spectral picture.

$b_k = \langle b, \phi_k \rangle_{L^2(P)}$. This represents the squared error from failing to capture the $k$-th component of the true function.

If the student does include mode $k$ in its active set $S_S$ (i.e., $\lambda_{S,k} > 0$), the situation depends on whether $k \in S_W$.

- If mode $k$ is active for the student ($k \in S_S$) but not for the teacher, then the situation is the same as not including $k$, with regards to the influence on $ER(\hat{f})$.

- If mode $k$ is active for both the student ($k \in S_S$) and the teacher ($k \in S_W$), the contribution to the squared bias from this mode is $b_k^2$. The contribution to variance from this mode is $\sigma^2/\lambda_{S,k}$. Thus, the total error contribution from this mode is $b_k^2 + \sigma^2/\lambda_{S,k}$.

Therefore, we only need to decide for a mode $k$ where the teacher provides a signal (i.e., $k \in S_W$). the student should choose to include this mode in its active set $S_S$ if its contribution to error when included is less than its contribution if dropped:

$$b_k^2 + \frac{\sigma^2}{\lambda_{S,k}} < (f_k^* + b_k)^2.$$

## 5.5 Intuitive Understanding and Mode-selection

The formal bias–variance expression in equation 1 can be understood geometrically. A convenient way is to visualize the eigen–axes of the kernel operator as orthogonal "mode" directions. The teacher's mean signal occupies a subset $S_W$ of these axes, with bias coefficients $b_k$ and variance $\sigma^2$ concentrated on the same span. The student chooses a subset $S_S$, and its error along each shared axis decomposes into a deterministic term $b_k^2$ and a stochastic term $\sigma^2/\lambda_{S,k}$. If a mode is ignored ($k \notin S_S$) the student pays $(f_k^* + b_k)^2$; if it is kept the student pays $b_k^2$ plus an inverse–eigenvalue variance. Figure 5.4 plots these ingredients as $m = |S_S|$ expands left-to-right: at first the student enjoys a large bias drop with minimal variance growth, but beyond the shared signal span variance dominates and the risk eventually exceeds the teacher's.

In reality, we can apply three simple heuristics when designing a kernel student to learn from a weak teacher.

1. Match but do not exceed the teacher's reliable subspace. First, estimate which low-frequency directions (typically those with large eigenvalues) carry the teacher's useful signal regarding $f_*$, rather than primarily its bias or noise. This estimation might draw on preliminary spectral analysis of the teacher's outputs or leverage domain knowledge. It is critical to initially restrict the student kernel to this identified reliable span. Expanding $S_S$ prematurely risks incorporating modes where the teacher's information is misleading or excessively noisy, which can impair generalization. Overestimating this subspace invites unnecessary variance and bias adoption from the teacher, while underestimation may omit crucial components of $f_*$.

2. Exploit eigenvalue decay for variance control. The student kernel's inverse eigenvalues, $1/\lambda_{S,k}$, scale the impact of teacher noise. Kernels with rapidly decaying eigenvalue spectra (e.g., squared-exponential or high-smoothness Matérn kernels) naturally attenuate noise in higher-frequency modes because their $\lambda_{S,k}$ values diminish quickly. Such a kernel allows the student to tolerate a slightly larger set of active modes $S_S$—potentially including more higher-frequency modes from the teacher—than a kernel with a flatter spectrum (like a linear kernel, or the $\lambda_{S,k} = 1$ idealization in our experiments). This can afford greater bias reduction by incorporating more of $f_T$'s signal, while the sharply decreasing eigenvalues help manage the cumulative variance cost. Conversely, if the spectrum lacks aggressive decay, the variance penalty for each additional mode can escalate rapidly, making an expanded $S_S$ less viable.

3. Validate with a tiny held-out set of true labels ($f_*$). Although abundant true labels would negate the W2S premise, even a very small "gold-standard" validation set offers significant practical value. Such a limited set can suffice to empirically track the student's error as modes are incrementally added to $S_S$. Crucially, it aids in pinpointing the U-shaped risk curve's upturn, which signals that adding more modes is counterproductive, leading to increased total error from overwhelming variance or adopted teacher bias. This facilitates early stopping in mode selection, thereby preventing the student from excessively incorporating the teacher's errors and noise. This approach is particularly useful because direct estimation of $b_k$ or precise spectral properties is often not feasible.

Though our theoretical work centers on kernel ridgeless regression, the findings about how spectral properties and modal selection influence the bias-variance trade-off are also useful for understanding W2S generalization in neural networks. This connection is clearest in lazy training or Neural Tangent Kernel (NTK) regimes, where sufficiently wide neural networks behave much like kernel machines. A two-layer random feature model under NTK dynamics, for example, performs KRR. The effective kernel spectra of these neural networks often decay more rapidly than the idealized flat spectrum considered in our KRR experiments in Section 6. Such faster decay provides neural networks with a better ability to reduce variance from less reliable or higher-frequency teacher modes. Consequently, these networks can engage a broader set of modes to learn complex signals and reduce bias from the teacher, while the steeper spectral fall-off helps control the associated variance cost. This places them advantageously within the W2S regime depicted by the U-shaped risk curve (Figure 5.4). Beyond these fixed-kernel analogies, the feature-learning capabilities of deep networks also equip them to develop internal representations specifically tailored to disentangle the true underlying signal from a teacher's characteristic imperfections, further bolstering their W2S potential.

# 6 EXPERIMENTS

To concretely explore the efficacy of the proposed student model selection strategy, we conduct a series of simple, synthetic, small-scale experiments.

## 6.1 SYNTHETIC EXPERIMENT

To illustrate the bias–variance trade–off predicted by equation 1, we constructed a controlled one–dimensional regression task. The ground–truth function is a smooth low–frequency signal $f_\star(x) = \cos(2\pi x) + 0.5\cos(4\pi x)$, $x \in [0, 1]$. A weak teacher provides labels $y = f_\star(x) + b(x) + \xi$, where $b(x) = \sum_{k=6}^{10} 0.4\cos(2\pi kx)$ injects high–frequency bias and $\xi \sim \mathcal{N}(0, \sigma^2)$ with $\sigma^2 = 0.04$ adds noise that is spectrally restricted to the same modes $k \in S_W = \{1, 2, 6, \ldots, 10\}$. A strong student is a truncated cosine series $f_S^{(m)}(x) = \sum_{k=1}^{m} \hat{a}_k \cos(2\pi kx)$ fitted by least squares to $n = 2\,000$ teacher–generated samples. Hence the student's kernel has eigenvalues $\lambda_{S,k} = 1$ on the first $m$ cosine modes and 0 elsewhere. We swept $m = 1, \ldots, 10$ and estimated the test MSE on $10\,000$ fresh points for each of them, averaging over 50 Monte Carlo runs.

Figure 6.1 plots: (i) the empirical test MSE of the student (orange dots), (ii) the theoretical prediction $\text{Bias}^2 + \text{Var}$ from equation 1 (black dashed), and (iii) the teacher's population error $\|f_T - f_\star\|_{L^2}^2 + \sigma^2$ (red line). For small truncation levels $m = 1$–$2$ the student keeps only the two low–frequency modes shared with the teacher, thereby filtering out the detrimental high–frequency bias. The variance term involves just the shared modes, so the overall error falls well below the teacher's line and shows W2S

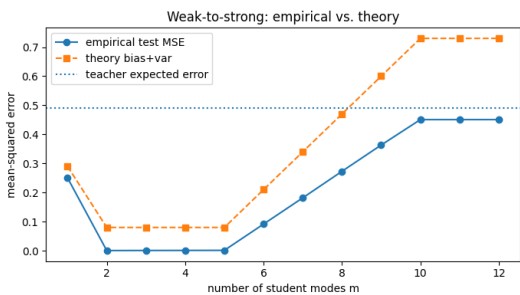

Figure 2: Empirical and theoretical MSE vs number of student modes in synthetic experiment.

generalization. As $m$ increases further, the student begins to admit the biased modes $k \geq 6$. While this reduces bias, it simultaneously incorporates additional noise contributions $\sigma^2/\lambda_{S,k}$, causing the MSE to rise sharply. The resulting U–shaped curve predicted by theory matches the empirical measurements, showing the trade–off in selecting the student's modal support $S_S$.

## 6.2 Experiment in CIFAR-10

Figure 6.2 shows a controlled weak-to-strong experiment carried out on the cats-versus-dogs subset of CIFAR-10. All 12 000 images of the two classes were flattened into 3 072 dimensional vectors and then centered and whitened. A principal component basis was extracted once from the full data matrix and served as the common functional modes for both teacher and student. A synthetic regression target, intended to stand in for the ground–truth function $f_\star$, was constructed so that it lived entirely in the first five principal components. A weak teacher was then obtained by adding two kinds of imperfection to that target: (i) a genuine bias term that occupies only modes 6-8 and (ii) independent Gaussian noise with variance $\sigma^2$. By design, this teacher has mean squared error of about $0.37$ when measured against the true target.

Students of increasing capacity were created by fitting ordinary least squares models that are allowed to use only the first $m$ principal components, with $m$ ranging from 1 to 20. The blue curve reports their empirical mean squared error on a disjoint test split, while the red dashed curve reports the theoretical risk given by equation 1, evaluated exactly from the known bias and variance terms under the simplifying assumption $\lambda_{S,k} = 1$.

When only one or two modes are kept the student cannot capture the low frequency content of $f_\star$, so its error is dominated by squared bias and starts high. As soon as $m$ reaches five the student is expressive enough to match every component of the signal while still ignoring the biased higher frequency modes, and the empirical error falls by two orders of magnitude, comfortably beating the teacher. Beyond that point additional modes invite both the teacher's bias and its label noise into the student's hypothesis space. The variance term in equation 1 grows linearly with $m$ and the empirical curve climbs until it settles on a broad plateau.

Although the theoretical curve is not numerically exact, since its variance term is slightly optimistic because real eigenvalues differ from $\lambda_{S,k} = 1$, it reproduces the qualitative behavior of the empirical results. There is a steep descent driven by bias reduction, a pronounced minimum when the informative modes are all present but biased modes are still excluded, and a subsequent rise once those biased modes become available. Over the interval $m \in [4, 5]$ both curves lie below the weak–teacher line, showing a region of weak-to-strong generalization in which a modestly sized student outperforms the source of its supervision.

## 7 Limitations and Future Work

Our theoretical investigation primarily analyzed error decompositions and W2S conditions in an asymptotic regime where the number of samples $n$ approaches infinity, meaning practical finite-sample performance may naturally differ from these idealized limits. The conceptual derivations for optimal student mode selection often assumed access to true spectral properties of the underlying

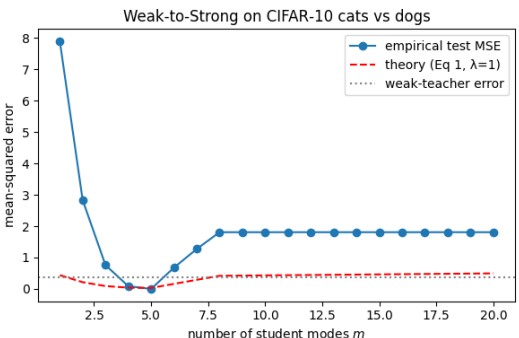

Figure 3: Empirical and theoretical MSE vs number of student modes in CIFAR-10 experiment. A correct choice of modes in student model is key to realization of weak-to-strong generalization.

functions and the kernel, which are typically unknown in real-world scenarios. Furthermore, this study focused specifically on a KRR student model operating under an assumption of i.i.d. noise from teacher variability, and other student architectures or more complex error structures could exhibit different W2S dynamics.

Consequently, the proposed heuristics for student mode selection, while illustrative of ideal trade-offs, faces practical implementation challenges due to its reliance on these unknown quantities and the potential difficulty or cost of acquiring true validation labels for $f_*$. While this work identifies key asymptotic error components and conditions for W2S within our KRR framework, it does not provide rigorous, non-asymptotic performance bounds for finite $n$. Deriving such finite-sample guarantees, which would explicitly detail the dependence on sample size and model complexities, remains an important direction for future research to fully bridge the current theoretical insights with practical W2S applications.

The kernel view in this work clarifies why variance suppression is hard in generic feature spaces: the student must sit on a vanishing tail of large eigenvalues. This insight motivates exploring non-stationary kernels whose spectrum is adapted to the teacher's error profile. Another promising direction is to pair the spectral filter with a learned noise model that down-weights teacher labels in high-variance regions. Finally, translating our analysis to deep neural students requires extending recent random feature proofs to the mismatched teacher–student regime, a pursuit we leave for subsequent work.

## 8    CONCLUSION

This paper presented a spectral analysis of a kernel ridgeless regression (KRR) student learning from a weak teacher characterized by systematic bias and output variance, seeking to deepen the understanding of Weak-to-Strong (W2S) generalization. Our derived asymptotic bias-variance decomposition indicates that successful W2S generalization is closely linked to the student's kernel. Specifically, outperformance can involve favorably reconfiguring inherited teacher bias, where operational modes strategically filter biased components, or substantially suppressing teacher output variance, a capability related to its kernel's eigenvalue spectrum. The student's kernel choice is thus highlighted as important for navigating this bias-variance trade-off, complementing considerations of mere model capacity. These theoretical insights, supported by numerical experiments, suggest that further W2S advancements may benefit from exploring how student models can leverage such identified spectral and modal properties to better manage learning from flawed supervision.

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

## A  DERIVATION OF BIAS AND VARIANCE COMPONENTS FOR THE KRR STUDENT

This appendix provides a detailed derivation of the bias and variance components for the ridgeless Kernel Ridge Regression (KRR) student estimator, $\hat{f}(x)$. These components are first derived conditional on a finite training set of inputs $X = \{x_j\}_{j=1}^n$, and then their asymptotic behavior is discussed. We operate under the assumptions outlined in Section 4 of the main text.

Recall the student is trained on labels $y_j = f_W(x_j) = f_T(x_j) + \xi_j$. The teacher's mean signal is $f_T(x) = \mathbb{E}[f_W(x)] = f_*(x) + b(x)$, and $\xi_j = f_W(x_j) - f_T(x_j)$ is the teacher's output variability with integrated variance $\sigma^2 = \mathrm{Var}(f_W)$. The student's kernel is $K_S(x, x') = \sum_{k \in S_S} \lambda_{S,k} \phi_k(x) \phi_k(x')$. The student estimator is $\hat{f}(\cdot|X, y) = S_n^*(X) K_n^\dagger(X) y$. Our goal is to analyze the components of the expected error $ER(\hat{f}) = \mathbb{E}_{X,\xi}[\|\hat{f} - f_*\|_{L^2(P)}^2]$.

### A.1  AVERAGE STUDENT ESTIMATOR (CONDITIONAL ON $X$)

First, we compute the expectation of the student estimator $\hat{f}$ over the noise $\xi$, conditional on a fixed set of training inputs $X$. Let $f_T(X) = [f_T(x_1), \ldots, f_T(x_n)]^T$.

$$
\begin{aligned}
\hat{f}_0(x|X) &= \mathbb{E}_\xi[\hat{f}(x|X, y)] \\
&= \mathbb{E}_\xi[S_n^*(X) K_n^\dagger(X) y] \\
&= S_n^*(X) K_n^\dagger(X) \mathbb{E}_\xi[y] \\
&= S_n^*(X) K_n^\dagger(X) (f_T(X) + \mathbb{E}_\xi[\xi])
\end{aligned}
$$

Since $\mathbb{E}_\xi[\xi] = 0$, we have:

$$
\hat{f}_0(x|X) = S_n^*(X) K_n^\dagger(X) f_T(X)
$$

Using the definition $S_n^*(X)\alpha = \sum_{j=1}^n \alpha_j K_S(x_j, x)$ and the Mercer decomposition $K_S(x_j, x) = \sum_{k \in S_S} \lambda_{S,k} \phi_k(x_j) \phi_k(x)$:

$$
\begin{aligned}
\hat{f}_0(x|X) &= \sum_{j=1}^n (K_n^\dagger(X) f_T(X))_j K_S(x_j, x) \\
&= \sum_{j=1}^n (K_n^\dagger(X) f_T(X))_j \left( \sum_{k \in S_S} \lambda_{S,k} \phi_k(x_j) \phi_k(x) \right) \\
&= \sum_{k \in S_S} \lambda_{S,k} \left( \sum_{j=1}^n (K_n^\dagger(X) f_T(X))_j \phi_k(x_j) \right) \phi_k(x)
\end{aligned}
$$

Let $\Phi_k(X) = [\phi_k(x_1), \ldots, \phi_k(x_n)]^T$. Then the term in the parenthesis is $\Phi_k(X)^T (K_n^\dagger(X) f_T(X))$. So,

$$
\hat{f}_0(x|X) = \sum_{k \in S_S} \lambda_{S,k} \left( \Phi_k(X)^T K_n^\dagger(X) f_T(X) \right) \phi_k(x)
$$

This is the student's prediction averaged over label noise, for a fixed set of training inputs $X$.

### A.2  CONDITIONAL BIAS TERM

The squared bias, conditional on $X$, of the student estimator with respect to $f_*(x)$ is:

$$
\mathrm{Bias}_*^2(\hat{f}|X) = \|\hat{f}_0(\cdot|X) - f_*\|_{L^2(P)}^2
$$

Substituting the expansion for $\hat{f}_0(\cdot|X)$ and $f_*(x) = \sum_k \langle f_*, \phi_k \rangle_{L^2(P)} \phi_k(x)$, and using the $L^2(P)$-orthonormality of $\{\phi_k\}$:

$$
\mathrm{Bias}_*^2(\hat{f}|X) = \sum_{k \in S_S} \left( \lambda_{S,k} (\Phi_k(X)^T K_n^\dagger(X) f_T(X)) - \langle f_*, \phi_k \rangle_{L^2(P)} \right)^2 + \sum_{k \notin S_S} \left( 0 - \langle f_*, \phi_k \rangle_{L^2(P)} \right)^2
$$

The second sum accounts for components of $f_*$ that are outside the student's kernel support $S_S$. This term represents the systematic error of the noise-averaged student estimator for a given $X$. It depends on how well $f_T(X)$ (which includes $f_*(X)$ and $b(X)$) can be reconstructed as $f_*$ by the student's finite-sample KRR procedure.

### A.3 Conditional Variance Term

The variance of $\hat{f}$ around its mean $\hat{f}_0$, conditional on $X$, due to the noise $\xi$ is:

$$\text{Var}_\xi(\hat{f}|X) = \mathbb{E}_\xi\left[\|\hat{f}(\cdot|X, y) - \hat{f}_0(\cdot|X)\|^2_{L^2(P)}\right]$$

We have $\hat{f}(\cdot|X, y) - \hat{f}_0(\cdot|X) = S_n^*(X)K_n^\dagger(X)y - S_n^*(X)K_n^\dagger(X)f_T(X) = S_n^*(X)K_n^\dagger(X)\xi$. Let $A(X) = S_n^*(X)K_n^\dagger(X)$. This is a linear operator mapping $\xi \in \mathbb{R}^n$ to a function in $L^2(P)$.

$$(A(X)\xi)(x) = \sum_{j=1}^n (K_n^\dagger(X)\xi)_j K_S(x_j, x)$$

We need to compute $\mathbb{E}_\xi[\|A(X)\xi\|^2_{L^2(P)}]$.

$$\|A(X)\xi\|^2_{L^2(P)} = \left\langle \sum_{j=1}^n (K_n^\dagger(X)\xi)_j K_S(x_j, \cdot), \sum_{l=1}^n (K_n^\dagger(X)\xi)_l K_S(x_l, \cdot) \right\rangle_{L^2(P)}$$

$$= \sum_{j=1}^n \sum_{l=1}^n (K_n^\dagger(X)\xi)_j (K_n^\dagger(X)\xi)_l \langle K_S(x_j, \cdot), K_S(x_l, \cdot)\rangle_{L^2(P)}$$

Let $M(X)$ be an $n \times n$ matrix with entries $(M(X))_{jl} = \langle K_S(x_j, \cdot), K_S(x_l, \cdot)\rangle_{L^2(P)}$. Using the Mercer decomposition for $K_S$:

$$(M(X))_{jl} = \left\langle \sum_{s \in S_S} \lambda_{S,s}\phi_s(x_j)\phi_s(\cdot), \sum_{t \in S_S} \lambda_{S,t}\phi_t(x_l)\phi_t(\cdot) \right\rangle_{L^2(P)} = \sum_{s \in S_S} \lambda_{S,s}^2\phi_s(x_j)\phi_s(x_l)$$

So, $\|A(X)\xi\|^2_{L^2(P)} = (K_n^\dagger(X)\xi)^T M(X)(K_n^\dagger(X)\xi) = \xi^T (K_n^\dagger(X))^T M(X) K_n^\dagger(X)\xi$. Since $K_n(X)$ is symmetric, $K_n^\dagger(X)$ is also symmetric. Let $B(X) = K_n^\dagger(X)M(X)K_n^\dagger(X)$, which is symmetric. The variance is $\mathbb{E}_\xi[\xi^T B(X)\xi]$. Given $\mathbb{E}[\xi] = 0$ and $\mathbb{E}[\xi\xi^T] = \sigma^2 I_n$:

$$\text{Var}_\xi(\hat{f}|X) = \sigma^2 \text{Tr}(B(X)) = \sigma^2 \text{Tr}(K_n^\dagger(X)M(X)K_n^\dagger(X))$$

Since $\text{Tr}(CD) = \text{Tr}(DC)$ for compatible matrices:

$$\text{Var}_\xi(\hat{f}|X) = \sigma^2 \text{Tr}(M(X)(K_n^\dagger(X))^2)$$

This formula gives the integrated variance conditional on the training inputs $X$. Substituting the sum form for $M(X)$:

$$\text{Var}_\xi(\hat{f}|X) = \sigma^2 \text{Tr}\left(\left(\sum_{s \in S_S} \lambda_{S,s}^2\Phi_s(X)\Phi_s(X)^T\right)(K_n^\dagger(X))^2\right)$$

$$= \sigma^2 \sum_{s \in S_S} \lambda_{S,s}^2 \text{Tr}(\Phi_s(X)\Phi_s(X)^T(K_n^\dagger(X))^2)$$

$$= \sigma^2 \sum_{s \in S_S} \lambda_{S,s}^2(\Phi_s(X)^T(K_n^\dagger(X))^2\Phi_s(X)).$$

This expression explicitly shows the dependence on the population eigenvalues $\lambda_{S,s}$, the population eigenfunctions evaluated at sample points $\Phi_s(X)$, the empirical Gram matrix $K_n(X)$, and the teacher's output variance $\sigma^2$.

### A.4 Asymptotic Analysis and W2S Conditions

The average (over $X$) of these finite-$n$ conditional bias and variance terms then leads to the asymptotic error. As $n \to \infty$, we discussed that $\mathbb{E}_X[\text{Bias}^2_*(\hat{f}|X)] \to \|P_{S_S}(f_T) - f_*\|^2_{L^2(P)}$ and $\mathbb{E}_X[\text{Var}_\xi(\hat{f}|X)] \to \sigma^2 \sum_{k \in S_S \cap S_W} \lambda_{S,k}^{-1}$ under suitable assumptions. The subsequent W2S conditions discussed in Section **??** of the main text then follow from these asymptotic limits.

