# OpenReview forum: "Understanding Weak-to-Strong Generalization: A Spectral Analysis"
_ICLR.cc/2026/Conference — ICLR 2026 Conference Withdrawn Submission_

### Official Review · Reviewer_HeQ1 · 2025-10-21

**Soundness:** 2
**Presentation:** 3
**Contribution:** 2
**Rating:** 2
**Confidence:** 3

**Summary:**

Authors investigate weak-to-strong generalization in a simplified kernel setting, where a kernel student fits labels generated by a biased teacher. They characterize the asymptotic (population) mean squared error with respect to the unbiased target function, and discuss the bias-variance tradeoff, which in particular depends on the match between the student eigenmodes and the modes in which lie the bias and signals. The qualitative finding is then illustrated in numerical experiment, in a synthetic setting, and on CIFAR data with a synthetic target.

**Strengths:**

The paper presents a particularly simple setting in which weak-to-strong generalization can arise, and present a fine study of the factors that can enter. This allows them to give some qualitative guidelines regarding the design of a good student kernel for a given task. The exposition of the ideas is sufficiently clear, and I found the intuitions well discussed.

**Weaknesses:**

I have numerous questions, which I list below, and would be willing to increase my score if these are clarified. The main weakness of the paper is that it is restricted to population kernel regression, with very limited insights that can be carried to more complex settings, and to the best of my reading, it presents very limited technical contribution or novelty. The main conclusions of the paper in terms of mode selection design for the student are also very intuitive. In light of this, I am for now in favor of rejection.

**Questions:**

One point of confusion is the definition of the noise. I believe the paper would greatly benefit from including further discussion, and an explicit definition of $\xi$, its decomposition on the $\phi_k$  basis, and the statistics of its components in this decomposition. In the current version, I am confused by:
- In assumption A 4.2, is $\xi_j$ denoting $\xi(x_j)$ ? Does its expectation and variance not depend on $x_j$ at all ?
- Similarly, in l. 368 : "$\xi\sim\mathcal{N}(0,\sigma^2)$ adds noise that is spectrally restricted to the same modes k=1,2..." I am confused. If $\xi\sim\mathcal{N}(0,\sigma^2)$ truly is a Gaussian additive noise, I believe it adds noise only to mode k=0?

Other questions:
- l.183: to the best of my reading, $S*$ is undefined?
- l. 260: Doesn't a small intersection $S_S\cap S_W$ also imply a higher bias ? Is there a tradeoff?
- l. 221: I am confused about the first term, and notably why it involves $f_*+b$ instead of just $f_*$? Can the authors clarify or add further discussion ? For instance, if $S_S\cap S_W=S_S$, wouldn't $f_{S_S\cap S_W}$ be zero on the modes in $S_W\setminus S_S$?
- In Fig.2, I am confused as to why the theory curve (which I believe is asymptotic) corresponds to higher error compared to a finite $n$ curve ?

---

### Official Review · Reviewer_fjij · 2025-10-28

**Soundness:** 2
**Presentation:** 2
**Contribution:** 1
**Rating:** 2
**Confidence:** 4

**Summary:**

The paper studies generalization performance of kernel regression from a spectral perspective. It derives a spectral bias-variance model risk decomposition and accordingly discusses strategies to minimize the test error. Some experiments are provided.

**Strengths:**

Unfortunately, I don't see much in terms of strengths in this submission. This is a fairly basic study that also has significant issues in exposition and math (see Weaknesses below). As positive aspects, the addressed problem is reasonable, the text is on the whole readable; the paper provides a literature review and experiments with a synthetic and a realistic datasets.

**Weaknesses:**

**Wrong statements / math errors**

I don't understand at all the formula in line 231 that claims a decomposition of the error variance as the sum of inverted eigenvalues of the kernel operator. This formula is not proved anywhere in the paper and seems just wrong to me. It suggests, in particular, that we can reduce the error variance by any factor $a$ simply by rescaling the kernel by this factor $a$ (since this also rescales the eigenvalues of the kernel operator). But rescaling the kernel does not, in fact, change the ridgeless regression model (and for ridge models with a positive regularization this would only rescale the regularization coefficient).

Moreover, it is unclear why the variance would include just the coefficient $\sigma^2$ characterizing the point noise, and not more detailed spectral properties of the noise. The paper is obscure regarding precise noise assumptions, but it seems to have a nontrivial correlation structure (line 158: "*The noise ξ(x) is also assumed to have its spectral content primarily within $S_W$.*")

The subsequent exposition in the paper, in particular the key risk decomposition formula (1), substantially relies on the equation in line 231 and is rendered invalid by the invalidity of this equation.

**Lack of novelty/contribution**

The overall contribution of the paper is a fairly basic (and probably wrong) spectral risk decomposition formula (1) developed within a standard spectral RKHS framework, along with an associated free-form discussion of risk reduction strategies. This discussion is not accompanied by any new clearly formulated results, and essentially just lists some more or less elementary bias/variance reduction ideas.
I don't see any significant mathematical novelty or new important well-articulated practical takeaway messages in this work.

**Sloppy mathematical writing**

The mathematical writing is sloppy and poorly structured in general. In addition to the already mentioned wrong formula in line 231, there are many unclear or vague formulations. For example, very little is said about the noise (is it even Gaussian or not?), it is only assumed to have its spectral content "*primarily within $S_W$*". In line 646, convergence is claimed to hold "*under suitable assumptions*". It is also claimed that it was discussed, but I don't see where it was discussed. Appendix provides some derivations for formulas presented in the main text, but without clear references, and does not seem to cover all formulas.

**Questions:**

N/A

---

### Official Review · Reviewer_3FED · 2025-10-31

**Soundness:** 1
**Presentation:** 1
**Contribution:** 1
**Rating:** 2
**Confidence:** 4

**Summary:**

The paper studies W2S generalization through a kernel ridgeless regression student operating in an RKHS, learning from a teacher with systematic bias and output variance.
Using an RKHS spectral decomposition, the authors derive an asymptotic bias–variance decomposition of the student’s risk and identify mode-selection conditions under which the student outperforms the teacher.
The key theoretical result, Eq. (1) (line 236), expresses the student's population risk as the sum of three terms: (i) the error from modes outside the intersection of the teacher and student modes that is not learnable by the student via W2S, (ii) the bias in $k$ shared modes, $b_k^2$, and (iii) the variance in $k$ shared modes, $\sigma^2 / \lambda_{S,k}$, where $\lambda_{S,k}$ is the student's eigenvalue for mode $k$.
This leads to mode selection and variance reduction conditions for W2S: with $f_k$ being the ground-truth coefficient for mode $k$, (i) $b_k^2 + \sigma^2 / \lambda_{S,k} < (f_k + b_k)^2$ and (ii) $\sum_{k \in S_W \cap S_S} 1/\lambda_{S,k} < 1$.
The theoretical findings are supported by experiments on a 1D synthetic example and a PCA-based toy regression setup using CIFAR-10, which reveal a U-shaped risk curve as student capacity increases and highlight the importance of mode regularization in achieving W2S.

**Strengths:**

The topic of W2S generalization is timely and relevant. The abstract is well-written.

**Weaknesses:**

1. While the introduction provides good motivations for why ridgeless regression is a reasonable choice for studying overparametrized models, what is left unaddressed for using kernel ridgeless regression is why and when the kernel model is a good proxy for neural networks in the W2S setting. The authors may consider adding a discussion on this point to strengthen the motivation for their model choice.
2. The comparison with related works in Section 2 is not fundamental or rigorous, and the novelty of the theoretical results appears incremental.
W2S has been extensively analyzed in various linear/kernel regression settings, e.g., Ildiz et al. (2024), Wu & Sahai (2024), Xue et al. (2025), Dong et al. (2025), Medvedev et al. (2025), Moniri & Hassani (2025), Liu et al. (2025), among others. In the kernel/linear setting, mode pruning can be viewed as a form of regularization with known connections to common techniques such as ridge regression and early stopping.
At high level, the key message of this paper on mode selection in a kernel setting does not seem to be fundamentally different from the well-known message about the importance of regularization in W2S, highlighted empirically in the initial W2S paper, Burns et al. (2023), and theoretically analyzed in the aforementioned works from various perspectives.
3. The theoretical results in Sections 4 and 5 are disorganized, feel like casual notes rather than a polished manuscript. See the "Questions" section below for specific points.
3. The problem setup and assumptions are hand-wavy and inconsistent. For example, Assumption 4.2 defines per-sample i.i.d. noise $\xi(x) \in \mathbb{R}$ with mean $0$ and variance $\sigma^2$, while Assumption 3.2 restricts the "spectral content" of $\xi(x)$ primarily within $S_W$, where the term "spectral content" is not clearly defined, and a scalar i.i.d. noise process does not have a well-defined spectral support in the student's eigenbasis unless a functional noise process with a specified covariance operator is given.
5. The experiments on the 1D synthetic task and the toy PCA experiment using CIFAR-10 are too limited in scope and contrived in setting, even given the theoretical nature of the paper.

I am happy to rectify any misunderstandings of the paper and adjust my assessment if the authors can provide clarifications and counter-arguments for the above points.

Ildiz, M. E., Gozeten, H. A., Taga, E. O., Mondelli, M., & Oymak, S. (2024). High-dimensional analysis of knowledge distillation: Weak-to-strong generalization and scaling laws. arXiv preprint arXiv:2410.18837.

Moniri, B., & Hassani, H. (2025). On the Mechanisms of Weak-to-Strong Generalization: A Theoretical Perspective. arXiv preprint arXiv:2505.18346.

Medvedev M, Lyu K, Yu D, Arora S, Li Z, Srebro N. Weak-to-strong generalization even in random feature networks, provably. arXiv preprint arXiv:2503.02877. 2025 Mar 4.

Liu, C., Dong, Y., & Lei, Q. (2025). Does Weak-to-strong Generalization Happen under Spurious Correlations?. arXiv preprint arXiv:2509.24005.

Wu, D. X., & Sahai, A. (2024). Provable weak-to-strong generalization via benign overfitting. arXiv preprint arXiv:2410.04638.

Xue, Y., Li, J., & Mirzasoleiman, B. (2025). Representations shape weak-to-strong generalization: Theoretical insights and empirical predictions. arXiv preprint arXiv:2502.00620.

Burns, C., Izmailov, P., Kirchner, J. H., Baker, B., Gao, L., Aschenbrenner, L., ... & Wu, J. (2023). Weak-to-strong generalization: Eliciting strong capabilities with weak supervision. arXiv preprint arXiv:2312.09390.

**Questions:**

1. Why is the variance reduction criterion $\sum_{k \in S_W \cap S_S} 1/\lambda_{S,k} < 1$ sensitive to kernel scaling (i.e., multiplying the kernel by a constant factor)?
2. I think there is something wrong in the analysis in Appendix A.3, leading to the problem in Q1 above.

Below are some minor questions and comments noted during my reading:
1. Using "KRR" as the abbreviation for "kernel ridgeless regression" is uncommon. "KRR" typically refers to "kernel ridge regression" in the literature. Consider using a different abbreviation to avoid confusion.
2. Link error in line 647, Appendix A.4.

---

### Official Review · Reviewer_BMq1 · 2025-10-31

**Soundness:** 3
**Presentation:** 2
**Contribution:** 1
**Rating:** 2
**Confidence:** 3

**Summary:**

The paper study weak-to-strong generalization in kernel ridgeless regression via a spectral bias-variance decomposition. The analysis identifies two main routes for outperforming a weak teacher: bias filtering via mode selection and variance suppression exploiting eigenvalue decay. Synthetic toy experiments validate the theoretical expressions.

**Strengths:**

The paper adresses a relevant problem, connecting kernel theory to W2S generalization. The eposition is clear and complemented by intuitive insights and clear explanations. Experiment match the qualitative behaviour predicted by the theory.

**Weaknesses:**

1. The novelty claims are not convincingly supported. The theoretical analysis mostly relies on classical KRR tools, applied to the specific weak-to-strong framing. The paper does not derive new theorems, identify unexpected phenomena, or extend existing kernel theory in a significant way. Consequently, the contribution feels mostly incremental.

2. The experiments are restricted to toy examples, serving only as qualitative visualizations of the theory. Moreover, the theoretical framework does not directly inform practical settings: the proposed heuristics (e.g., estimating a “reliable subspace” or using a few labels for validation) are not sufficiently detailed and can contradict the core weak-to-strong premise.

3. All figure references point to sections rather than captions. Moreover, the description of Figure 2 (lines 374–377) does not correspond to the actual plot content, and the mismatch between the theoretical and empirical curves is not discussed in sufficient detail, especially regarding the sources of deviation.

4. Related-works are not adequately discussed, in particular the literature on kernel ridge regression.

These issues limit both the theoretical and practical impact of the paper. The analysis is technically correct but offers little novelty beyond established kernel theory and the empirical demonstrations do not convincingly validate or extend the proposed insights.

**Questions:**

1. I would like to ask the authors to emphasize the originality of your work in relations to the prior literature. What are the main conceptual and technical novelties?

2. Could you motivate further the motivation for focusing on ridgeless regression? Would regularization affects your conclusions?

3. What would be the main challenges in extending your analysis to the non-asymptotic limit, perhaps leveraging existing results (e.g. [1])?


[1] Misiakiewicz, Saeed. "A non-asymptotic theory of Kernel Ridge Regression: deterministic equivalents, test error, and GCV estimator"

---

### Note · Authors · 2025-11-16

I have read and agree with the venue's withdrawal policy on behalf of myself and my co-authors.